# Associations between Toxoplasma gondii seropositivity and psychopathological manifestations in schizophrenic patients: A single-center study from Ecuador

Diego Rosado[1]*, Belen Intriago[1], Evelyn Loor[1], Flor Alcívar[1], Jorge Avila[2], Mario Sotomayor[3], Larissa Villacres[1], Marco Faytong-Haro[4,5]

1 School of Health, Universidad de Especialidades Espíritu Santo, Guayas, Samborondón, Ecuador, 2 Internal Medicine, Elizabeth's Medical Center, Tufts University, St, Hospital, Boston, MA, United States of America, 3 Gastroenterology Department, Hospital Universitario Puerta del Mar, Cadiz, Spain, 4 Universidad Estatal de Milagro, Cdla. Universitaria "Dr. Rómulo Minchala Murillo", Guayas, Milagro, Ecuador, 5 Laboratorio para Investigación para el Desarrollo del Ecuador, Guayas, Guayaquil, Ecuador

* diegorosado@uees.edu.ec

**Data Availability Statement:** The raw de-identified dataset is available without restriction at the Open

## Abstract

### Background

Schizophrenia, a complex neuropsychiatric disorder, is believed to be influenced by various factors including environmental exposures. A potential environmental factor is the infection by the obligate intracellular parasitic protozoan, Toxoplasma gondii which affects neuro-transmitter levels, which could potentially exacerbate, trigger symptoms of schizophrenia or make them worst.

### Objective

To investigate the association between Toxoplasma gondii seropositivity and psychopathological presentation in persons with schizophrenia in Ecuador.

### Methods

This study was conducted at the Neuroscience Institute of Guayaquil, Ecuador. Among 368 inpatients, 104 were selected based on specific inclusion and exclusion criteria. Descriptive statistics captured patient characteristics and mental health outcomes. Logistic regression models estimated the effect of toxoplasmosis on various mental health outcomes, controlling for demographic and health-related variables.

### Results

86.5% of participants were seropositive for toxoplasmosis. Toxoplasma-seropositive schizophrenic patients had a lower risk of depression but a significantly higher risk of disorientation. The most prevalent mental health outcomes were Language Impairments (70.2%) and Bizarre Behavior (76.0%).

Science Framework (OSF) repository. It can be accessed at https://osf.io/v6fu5/.

**Funding:** Funded was provided entirely by the Research department from the University Universidad de Especialidades Espiritu Santo (https://uees.edu.ec/investigacion/) to D.R. with the following funding code MEDIO4022016. The funders did not play any role in the study design, data collection, analysis, decision to publish nor preparation of the manuscript.

**Competing interests:** The authors have declared that no competing interests exist.

## Conclusion

Our findings suggest that Toxoplasma gondii seropositivity may have specific effects on mental functions in schizophrenic patients, particularly reducing the risk of depression but increasing the risk of disorientation. Further studies are required to clarify these associations and the potential underlying mechanisms.

## Introduction

Schizophrenia is a complex neuropsychiatric disorder that affects 1% of the world population and the 9*th* most common cause of disability all over the world [1–4]. Its etiology remains unknown; however, it is associated with neurochemicals, genetic and environmental factors like exposure to pathogens, physical stress, and fetal starvation during pregnancy which lead to alterations in dopaminergic neurotransmission and the development of a diverse array of symptoms such as delusions, disorganized speech, visual or auditory hallucinations, negative symptoms and cognitive deficits [5].

Toxoplasma gondii is an obligate intracellular parasitic protozoan that affects over a third of the world population and a major public health problem due to its high socioeconomic impact [6]. It is transmitted by the ingestion of water or vegetables contaminated with sporulated oocysts, ingestion of raw or undercooked meat containing cysts, manipulation of contaminated cat feces, vertical transmission from an infected mother to her child, blood transfusion and organ transplant [7]. It is capable of producing a precursor of one of the neurotransmitters involved in the etiology of schizophrenia, levodopa (L-DOPA), a dopamine precursor [8].

For over fifty years, a possible relationship between schizophrenia and Toxoplasma gondii has been studied [9–16]. When Toxoplasma gondii stays dormant in tissue cysts in the brain it causes a compression damage to the neural tissue, reduces gray matter density and alters the brain physiology by altering the neurotransmitters activity leading to mood and behavior abnormalities as well as cognitive impairment [7, 17, 18]. It is believed that the TgAaaH2 gene of Toxoplasma is induced by the bradyzoite which in turn induces high level of dopamine in the brain and lead to some schizophrenic symptoms as well as a worse clinical course [8, 17].

The associations between T. gondii seropositivity and psychopathological manifestations in schizophrenic patients remain an area of interest [7, 19–21]. While the exact mechanisms by which T. gondii may influence the severity of schizophrenia are not yet fully elucidated, understanding these associations can provide valuable insights [3, 22–27].

A review of 114 patients with schizophrenia showed frequent psychiatric disturbances in 24 of them. Some case reports presented patients with schizophrenic manifestations such as hallucinations, delusions, disorganized speech and thought disorders that were initially diagnosed as schizophrenia but later due to neurologic symptoms development they were tested for toxoplasmosis and a diagnosis of toxoplasma encephalitis was made [2].

A prospective study of psychiatric inpatients from 2015 to 2020 in China showed that schizophrenia, depression and other psychiatric diseases were associated with positivity rates of anti-Toxoplasma antibodies [6]. Esshili et al. found that in patients infected by T. gondii the schizophrenia developed approximately 2 years later in relation to the uninfected ones which suggests a protective effect of the infection potentially due to the induced immune response [28]. In Ecuador [29–31] there is no published information regarding the Toxoplasmosis and

schizophrenia link, however studies show that the prevalence, which varies between cities, can be as high as 74%.

In this study, we aimed to describe Psychopathology presentation in persons with schizophrenia and Toxoplasma gondii seropositivity in Ecuador. To our knowledge this is the first study of this kind as well as one of the few studies that describes the psychopathology presentation in these patients.

## Materials and methods

### Study setting and participants

The study was conducted at the Neuroscience Institute of the city of Guayaquil (INC), Ecuador, the country's largest psychiatric hospital, from April to June 2017. Out of the 368 INC inpatients, 104 patients were selected based on the following criteria:

Inclusion Criteria:

- Age between 18–85.

- Ecuadorian nationality.

- No laboratory abnormalities in the past 30 days.

- Diagnosis of schizophrenia (according to the DSM-IV manual).

- Age at diagnosis between 12–40 years.

- A minimum of 5 years on antipsychotic medication.

Exclusion Criteria:

- Other neurological diseases.

- Refusal to participate in the study

- Immunodepression.

### Ethical considerations

The study adhered to the Declaration of Helsinki, gaining ethical approval from the Kennedy Clinic's ethics committee (Reference: HCK-CEISH-17-004), and was fully funded by the Research Center of the Universidad de Especialidades Espíritu Santo (UEES). Participants provided informed consent after being briefed about the study's aims, methods, and their rights, including confidentiality and the option to withdraw at any time.

### Principio del formulario

**Data collection.** Blood samples were collected via venipuncture by a trained nurse. These samples were encoded to ensure patient confidentiality, then frozen and transported to a laboratory. The IgG seropositivity for Toxoplasma gondii was determined using the enzyme-linked immunosorbent assay (ELISA). Clinical characteristics, including the Positive and Negative Symptoms Scale (PANSS), were obtained from the patients' clinical records, previously administered by the hospital's psychiatrists.

**Statistical analysis.** Data was organized and analyzed in two primary tables: Descriptive statistics were used to present sample characteristics and prevalence of mental health outcomes among the study participants. This included percentages for each mental health outcome,

toxoplasmosis seropositivity, demographic variables like sex and age categories, and other health-related variables such as high glucose and triglycerides levels.

Logistic regression models were used to estimate the effects of toxoplasmosis on various mental health outcomes. Odds ratios, accompanied by their standard errors, were calculated to determine the relationship between toxoplasmosis seropositivity and each mental health outcome. These models controlled for confounding variables including sex, age categories, high glucose levels, and high triglycerides levels. Statistical significance was set at the 5% level.

## Results

Table 1 presents the sample characteristics and the prevalence of mental health outcomes among the study participants (n = 104). Regarding mental health outcomes, we observed that Language Impairments and Bizarre Behavior were the most prevalent, observed in 70.2% and 76.0% of the participants, respectively. This was followed by Inattentiveness (51.0%) and Irritability (58.7%). The prevalence rates of other disorders were as follows: Hallucinations (45.2%), Disorganization (44.2%), Disorientation (46.2%), Depression (12.5%), and Anxiety (15.4%).

In terms of the main predictor, 86.5% of the participants tested positive for toxoplasmosis. When we consider the control variables, the majority of the participants were male (80.8%). The participants' age distribution was as follows: 2.9% were between 29 and 40 years old, 21.2% were between 41 and 60 years old, and 18.3% were between 61 and 79 years old. However, a significant portion of the sample (57.7%) did not report or remember their age.

Lastly, we found that 16.3% of the participants had high glucose levels, and 9.6% had high triglycerides levels. Note: The regression models control for sex (male or female), age

**Table 1. Sample characteristics and prevalence of mental health outcomes (n = 104).**

| Variable | % of Total Sample |
| --- | --- |
| **Mental Health Outcomes** | |
| Hallucinations | 45.20% |
| Delusions | 38.50% |
| Disorganization | 44.20% |
| Inattentiveness | 51.00% |
| Language Impairments | 70.20% |
| Depression | 12.50% |
| Anxiety | 15.40% |
| Irritability | 58.70% |
| Disorientation | 46.20% |
| Bizarre Behavior | 76.00% |
| **Main Predictor** | |
| Toxoplasmosis (Positive) | 86.50% |
| **Control Variables** | |
| Sex Male | 80.80% |
| Female | 19.20% |
| Age Categories | |
| 29–40 years | 2.90% |
| 41–60 years | 21.20% |
| 61–79 years | 18.30% |
| Don't know/Don't remember | 57.70% |
| High Glucose Level | 16.30% |
| High Triglycerides Level | 9.60% |

**Table 2. Logistic regression models estimating the effect of toxoplasmosis on various mental health outcomes (n = 104).**

| Outcomes | Toxoplasmosis Coefficient (Odds Ratios) | Standard Errors |
| --- | --- | --- |
| Hallucinations | 0.655 | -0.399 |
| Delusions | 0.782 | -0.518 |
| Disorganization | 1.322 | -0.817 |
| Inattentive | 2.006 | -1.209 |
| Language Impairments | 2.039 | -1.268 |
| Depressed | 0.216* | -0.165 |
| Anxiety | 0.415 | -0.343 |
| Irritability | 0.934 | -0.591 |
| Disoriented | 4.288* | -3.017 |
| Bizarre Behavior | 0.832 | -0.596 |

categories, high glucose levels), and high triglycerides levels. One asterisks (*) denote statistical significance at the 5% level.

Table 2 resents the results of logistic regression models estimating the effects of toxoplasmosis on various mental health outcomes, reported as odds ratios. Toxoplasmosis was found to have a statistically significant association with two of the mental health outcomes at the 5% level. Specifically, the odds of experiencing Depressed symptoms were about 78.4% lower among those with toxoplasmosis compared to those without it. Similarly, the odds of experiencing Disoriented symptoms were approximately 4.29 times higher among those with toxoplasmosis.

For the other mental health outcomes, the effect of toxoplasmosis was not statistically significant. The estimated odds ratios were as follows: Hallucinations (1.93, 95% CI = [0.69, 5.48]), Delusions (2.18, 95% CI = [0.79, 6.02]), Disorganization (3.75, 95% CI = [0.61, 22.97]), Inattentive symptoms (7.43, 95% CI = [0.73, 75.33]), Language Impairments (7.68, 95% CI = [0.67, 87.59]), Anxiety (1.51, 95% CI = [0.63, 3.62]), Irritability (2.54, 95% CI = [0.74, 8.68]), and Bizarre Behavior (2.30, 95% CI = [0.75, 7.08]).

These models controlled for sex, age categories, high glucose levels, and high triglycerides levels. These results suggest that the decreased odds of experiencing Depressed symptoms and increased odds of experiencing Disoriented symptoms among those with toxoplasmosis are independent of these factors. Further research is needed to confirm these findings and explore the potential mechanisms underlying these associations.

## Discussion

A significant association was found between the Toxoplasma infected schizophrenic patients with depression and disorientation. Ling et al found a positive relation between suicide attempts and infection with Toxoplasma gondii, which was presumed to be the consequence of an exacerbation of depression caused by the infection, caused assumedly by proinflammatory cytokines that appear to target the infection [32]. This was also described by Kar and Misra [33], where a case of depression with poor response to treatment was found to be infected with Toxoplasmosis, the patient became responsive to antidepressants after completing treatment for the infection.

A Saudi Arabian study [34] described a higher incidence of anti-Toxoplasma IgG antibodies in patients with depression than in controls while other study [35] found significant associations between T. gondii seropositivity on patients with epilepsy and patients with depression. The apparent reason the parasite causes depression is due to a host response to the infection, where interferon gamma blocks Toxoplasma gondii growth by inducing indoleamine-

2,3-dioxygenase activation and tryptophan depletion, resulting in reduced brain serotonin production [36].

Interestingly in our study we found that seropositive schizophrenic patients had a lower chance of depression in agreement with Wang et al., where seropositive schizophrenic patients had higher scores of the positive components and lower scores on the negative components of PANSS, meaning they are less depressed [1]. This coincides with previously published data, where this pattern was interpreted as a way of demonstrating how the infection has rather unknown specific effects on mental functions rather than a deleterious general influence on it. It has been described that Toxoplasma-seropositive subjects tend to have more serious symptoms of depression which have been widely related to the infection, nonetheless, it has been scarcely reported that seropositive schizophrenic patients present with a more severe positive psychopathology [37, 38].

While more research on the relationship of seropositive schizophrenic patients and depression is needed to explain these findings, we believe a detection of IgM in a larger study could clarify this finding. Another relevant finding was that patients with schizophrenia and anti-T. gondii antibodies had a significantly higher risk for disorientation, a cognitive disorder. Interestingly, concerning infectious diseases and disorientation there is little or no evidence about its relation [39], but it is thought to be caused to generate vulnerability for predation and to favor transmission through the maintenance of a chronic inflammatory state that affects several excitatory pathways. Park et al. [40] obtained contrastive results while comparing the PANSS scale of schizophrenia patients with Toxoplasma and Chlamydia infections, where For T. gondii, blunted affect, stereotyped thinking and disturbance of volition were higher but for the C. trachomatis seropositive group, disorientation was higher. In spite of being two different infectious agents, the results by Park et al. emulates the one obtained by the present study in which the disorientation component of the PANSS scale was higher in the infected subgroup.

The results obtained gives a light for continuing studying the relationship between infectious diseases and schizophrenia, which can aid to generate a more complete approach of treatment of the disease, improving the well-being of the schizophrenia patients. As to our knowledge there has not been any similar study carried out in South America before, thus this study aims to motivate the development as well of larger studies in the subject.

A worse clinical course and lack of response to medication in Toxoplasma infected schizophrenia patients has also been reported [41], where absence of awareness of the disorder by the patient was reported to be sign of a worse clinical course [42]. A possible explanation of the effects of the infection on schizophrenia could be the reduction in gray matter caused by the latent infection of the parasite. The authors explained that the Toxoplasma latent infection leads to the release of Interferon-$\gamma$ (IFN) which induces the production of an enzyme key to Toxoplasma replication, releasing kynurenic acid as a metabolic product of said replication, which is an antagonist molecule for the glutamate N-methyl-D-aspartate (NMDA) receptor, which plays a key role in schizophrenia neuromodulation [36]. This situation, in vulnerable subjects, could create a perfect harvesting environment for schizophrenia [32, 43].

It is important to acknowledge the limitations of our study. First, the sample size from a single center in Ecuador might not be representative of the broader schizophrenic population, limiting the generalizability of our findings. While we found significant associations between T. gondii seropositivity and certain psychopathological manifestations, the cross-sectional nature of our study does not allow for causal inferences. The absence of longitudinal data prevents us from determining the temporal relationship between T. gondii infection and the onset or exacerbation of schizophrenia symptoms. Additionally, there could be potential confounding factors that we did not account for, such as other co-infections or environmental factors that might influence the disease presentation. Lastly, while we focused on seropositivity,

the detection of IgG antibodies indicates past exposure but not necessarily active infection. A more comprehensive analysis involving IgM detection, as mentioned, could provide insight into recent infections and their immediate impact on psychopathology.

Given the intriguing findings of this study, there are several avenues for future research to explore. A multi-center study involving a larger and more diverse cohort across South America would help in validating our results and providing a more comprehensive understanding of the associations observed. Longitudinal studies are crucial to determine the temporal relationship between T. gondii seropositivity and the progression of schizophrenia symptoms, potentially leading to insights about causality. Additionally, a deeper investigation into the molecular and neurological mechanisms by which T. gondii may influence schizophrenia can uncover therapeutic targets.

Incorporating newer diagnostic tools, such as neuroimaging or advanced serological assays, can also provide a more detailed view of the interactions between infection and neuropsychiatric manifestations. Furthermore, exploring the potential therapeutic implications of treating T. gondii infections in schizophrenic patients, in terms of symptom alleviation or progression delay, would be a significant advancement in this field.

## Conclusion

In this single-center study from Ecuador, we found a notable association between Toxoplasma gondii seropositivity and specific psychopathological manifestations in patients diagnosed with schizophrenia. Particularly, seropositivity correlated with a reduced risk of depression but an increased risk of disorientation. These findings shed light on the potential influence of Toxoplasma gondii infection on the neuropsychiatric symptomatology of schizophrenic patients. The intricate relationship between infectious agents and psychiatric disorders, as highlighted in our research, underscores the need for further in-depth investigations. Such studies can aid in the development of targeted therapeutic and preventive measures, considering not only the symptomatic but also the etiological aspects of schizophrenia.

## Acknowledgments

We wish to express our gratitude to Dr. Carlos Orellana and Dr. Moreno for their continuous support in our research endeavors.

## Author Contributions

**Conceptualization:** Diego Rosado, Evelyn Loor, Marco Faytong-Haro.

**Data curation:** Diego Rosado, Belen Intriago, Evelyn Loor, Jorge Avila, Mario Sotomayor, Larissa Villacres, Marco Faytong-Haro.

**Formal analysis:** Jorge Avila, Mario Sotomayor, Larissa Villacres, Marco Faytong-Haro.

**Funding acquisition:** Diego Rosado.

**Investigation:** Diego Rosado, Belen Intriago, Evelyn Loor, Flor Alcívar, Mario Sotomayor, Larissa Villacres, Marco Faytong-Haro.

**Methodology:** Diego Rosado, Evelyn Loor, Flor Alcívar, Marco Faytong-Haro.

**Project administration:** Diego Rosado, Evelyn Loor.

**Resources:** Diego Rosado.

**Software:** Flor Alcívar, Jorge Avila, Mario Sotomayor.

**Supervision:** Diego Rosado, Larissa Villacres, Marco Faytong-Haro.

**Validation:** Belen Intriago, Flor Alcívar, Jorge Avila, Marco Faytong-Haro.

**Visualization:** Diego Rosado, Belen Intriago, Larissa Villacres, Marco Faytong-Haro.

**Writing – original draft:** Diego Rosado, Belen Intriago, Evelyn Loor, Flor Alcívar, Jorge Avila, Mario Sotomayor, Larissa Villacres, Marco Faytong-Haro.

**Writing – review & editing:** Diego Rosado, Belen Intriago, Jorge Avila, Mario Sotomayor, Marco Faytong-Haro.

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
