## [Decision Letter · Decision Letter 0]

26 Dec 2023

PONE-D-23-33384Associations between Toxoplasma gondii Seropositivity and Psychopathological Manifestations in Schizophrenic Patients: A Single-Center Study from EcuadorPLOS ONE

Dear Dr. Rosado,

Thank you for submitting your manuscript to PLOS ONE. After careful consideration, we feel that it has merit but does not fully meet PLOS ONE’s publication criteria as it currently stands. Therefore, we invite you to submit a revised version of the manuscript that addresses the points raised during the review process.

We look forward to receiving your revised manuscript.

Kind regards,

Masoud Foroutan, Ph.D; Assistant Professor

Academic Editor

PLOS ONE

3. In the online submission form, you indicated that [The data underlying the results can be fully provided upon request.].

Reviewers' comments:

Reviewer's Responses to Questions

**Comments to the Author**

1. Is the manuscript technically sound, and do the data support the conclusions?

Reviewer #1: Yes

Reviewer #2: Yes

2. Has the statistical analysis been performed appropriately and rigorously? 

Reviewer #1: Yes

Reviewer #2: Yes

3. Have the authors made all data underlying the findings in their manuscript fully available?

Reviewer #1: Yes

Reviewer #2: Yes

4. Is the manuscript presented in an intelligible fashion and written in standard English?

Reviewer #1: Yes

Reviewer #2: Yes

5. Review Comments to the Author

Reviewer #1: The Rosado et el. manuscript describes very good the relation between Toxoplasma gondii infection and Psychopathological manifestation in Schizophrenic patients.

The language used is very correct and clearly used.The title is very well choosen and the abstract very good written. The statistical analysis were very performed. The conclusion are very good build.

Reviewer #2: The references exhibit inconsistency in formatting and fail to adhere to a unified citation style also lack recent scholarly sources, notably missing key studies from the past five years.

"The disconnect between the references cited within the text and their relevance to both the introduction and discussion sections of the article creates a significant discrepancy, undermining the coherence and integrity of the paper's argumentation."

6. PLOS authors have the option to publish the peer review history of their article (what does this mean?). If published, this will include your full peer review and any attached files.

Reviewer #1: No

Reviewer #2: No

---

## [Author Response · Author response to Decision Letter 0]

29 Dec 2023

Date: December 28th, 2023

To: Dr. Masoud Foroutan, Academic Editor, PLOS ONE

From: Dr. Diego Rosado

Subject: Rebuttal Letter - PLOS ONE Decision: Revision Required [PONE-D-23-33384]

Dear Dr. Foroutan,

I am writing to address the decision and the feedback provided for our manuscript, PONE-D-23-33384, titled "Associations between Toxoplasma gondii Seropositivity and Psychopathological Manifestations in Schizophrenic Patients: A Single-Center Study from Ecuador." This letter incorporates responses to both the reviewers' comments and the additional journal requirements outlined in your correspondence.

Response to Reviewers' Comments:

Reviewer #1 Comments:

1. The Rosado et el. manuscript describes very good the relation between Toxoplasma gondii infection and Psychopathological manifestation in Schizophrenic patients.

The language used is very correct and clearly used. The title is very well choosen and the abstract very good written. The statistical analysis were very performed. The conclusion are very good build.

Response: We are grateful for your positive feedback. We have maintained the manuscript's clarity and further refined it according to the journal requirements and Reviewer 1’s comments to better convey our paper.

Reviewer #2 Comments:

1. The references exhibit inconsistency in formatting and fail to adhere to a unified citation style also lack recent scholarly sources, notably missing key studies from the past five years.

2. The disconnect between the references cited within the text and their relevance to both the introduction and discussion sections of the article creates a significant discrepancy, undermining the coherence and integrity of the paper's argumentation.

Responses: 

1. We have updated the references to include key studies from the past five years (now more than 50% of references are from the past five years), ensuring our research is grounded in the most current and relevant scientific literature.

2. We have revised the entire list of references for consistency with PLOS ONE's citation style and included recent studies to enhance the manuscript's scholarly depth.

Responses to Journal Requirements:

Requirement 1: Manuscript Style and File Naming

Response: We have reviewed and applied the PLOS ONE style templates to our manuscript, ensuring full compliance with the formatting guidelines and file naming conventions.

Requirements 2 & 3: Data Depositing for Enhanced Citation and Accessibility

3. In the online submission form, you indicated that [The data underlying the results can be fully provided upon request.].

Response: In alignment with the journal's policy and to enhance citation potential, we have deposited our de-identified dataset in the Open Science Framework (OSF) repository. This action ensures our data is accessible and supports the integrity and reach of our work.

Requirement 4: Placement of Ethics Statement

Response: The ethics statement is now appropriately located solely in the Methods section of our manuscript, ensuring clarity and adherence to ethical guidelines.

In conclusion, we hope the revisions and responses provided address the concerns raised by the reviewers and meet the journal's requirements. We believe these changes significantly improve our manuscript and demonstrate our commitment to contributing valuable, accessible, and ethically sound research to the scientific community.

We appreciate the opportunity to refine our work and look forward to the possibility of our manuscript being published in PLOS ONE.

Thank you for considering our revised manuscript.

Kind regards,

Dr. Diego Rosado

---

## [Decision Letter · Decision Letter 1]

8 Jan 2024

Associations between Toxoplasma gondii Seropositivity and Psychopathological Manifestations in Schizophrenic Patients: A Single-Center Study from Ecuador

PONE-D-23-33384R1

Dear Dr. Rosado,

We’re pleased to inform you that your manuscript has been judged scientifically suitable for publication and will be formally accepted for publication once it meets all outstanding technical requirements.

Kind regards,

Masoud Foroutan, Ph.D; Assistant Professor

Academic Editor

PLOS ONE

Additional Editor Comments (optional):

Reviewers' comments:

Reviewer's Responses to Questions

**Comments to the Author**

1. If the authors have adequately addressed your comments raised in a previous round of review and you feel that this manuscript is now acceptable for publication, you may indicate that here to bypass the “Comments to the Author” section, enter your conflict of interest statement in the “Confidential to Editor” section, and submit your "Accept" recommendation.

Reviewer #2: All comments have been addressed

2. Is the manuscript technically sound, and do the data support the conclusions?

Reviewer #2: Yes

3. Has the statistical analysis been performed appropriately and rigorously? 

Reviewer #2: Yes

4. Have the authors made all data underlying the findings in their manuscript fully available?

Reviewer #2: Yes

5. Is the manuscript presented in an intelligible fashion and written in standard English?

Reviewer #2: Yes

6. Review Comments to the Author

Reviewer #2: (No Response)

7. PLOS authors have the option to publish the peer review history of their article (what does this mean?). If published, this will include your full peer review and any attached files.

Reviewer #2: **Yes: **Souzan Eassa

---

## [Editor Report · Acceptance letter]

15 Feb 2024

PONE-D-23-33384R1 

PLOS ONE

Dear Dr. Rosado, 

I'm pleased to inform you that your manuscript has been deemed suitable for publication in PLOS ONE. Congratulations! Your manuscript is now being handed over to our production team.

Kind regards, 

on behalf of

Dr. Masoud Foroutan 

Academic Editor

PLOS ONE